# Eleutheroside E supplementation prevents radiation-induced cognitive impairment and activates PKA signaling via gut microbiota

Chen Song[1,2,3], Fangyuan Duan[1,2,3], Ting Ju[1,2,3], Yue Qin[1,2,3], Deyong Zeng[1,2,3], Shan Shan[1,2,3], Yudong Shi[1,2,3], Yingchun Zhang[1,2,3] & Weihong Lu [1,2,3✉]

Radiation affects not only cognitive function but also gut microbiota. Eleutheroside E (EE), a principal active compound of Acanthopanax senticosus, has a certain protective effect on the nervous system. Here, we find a four-week EE supplementation to the $^{60}$Co-γ ray irradiated mice improves the cognition and spatial memory impairments along with the protection of hippocampal neurons, remodels the gut microbiota, especially changes of *Lactobacillus* and *Helicobacter,* and altered the microbial metabolites including neurotransmitters (GABA, NE, ACH, 5-HT) as well as their precursors. Furthermore, the fecal transplantation of EE donors verifies that EE alleviated cognition and spatial memory impairments, and activates the PKA/CREB/BDNF signaling via gut microbiota. Our findings provide insight into the mechanism of EE effect on the gut-brain axis and underpin a proposed therapeutic value of EE in cognitive and memory impairments induced by radiation.

[1] School of Chemistry and Chemical Engineering, Harbin Institute of Technology, Harbin, China. [2] Department of Food Science and Engineering, School of Medicine and Health, Harbin Institute of Technology, Harbin, China. [3] National and Local Joint Engineering Laboratory for Synthesis, Transformation, and Separation of Extreme Environmental Nutrients, Harbin Institute of Technology, Harbin 150001, China. ✉email: lwh@hit.edu.cn

With the advancement of science and technology, human beings have carried out a deeper exploration of space. The space environment is extremely complex including microgravity, radiation, extreme temperatures, and other extreme condition[1,2]. Irradiation can cause many pernicious effects, including cardiovascular disease, genetic mutations, and bone injure[3,4]. In addition, the hippocampus has shown as one of the touchy regions to be influenced by gamma rays, which promptly driven to significant modifications in cognitive capacities[5]. However, many anti-radiation agents are difficult to popularize because of their toxic side effects. Recently, there is a rising interest in how the microbiome influences the individual's generally reaction to radiation[6,7]. Many studies illustrated that the gut-brain axis establishes a two-way communication between the brain and the intestine[8,9]. It is a novel idea to search for a natural anti-radiation agent from the perspective of brain-gut axis. Indeed, the microbiota is able to affect the absorption and metabolism of drugs, bioactive compounds, or nutrients from food. On the other hand, drugs or bioactive compounds are able to shape the gut microbiota itself, where these changes may contribute to their pharmacologic profile[10]. More bidirectional communication pathways between the gut and brain need to be further investigated.

Acanthopanax senticosus (AS) is viewed as a medicinal and edible homologous plant broadly utilized in China[11]. The bioactivity of AS, including immune regulation, anti-radiation, and anti-oxidation, have stimulated people's intrigued[12,13]. Eleutheroside E (EE), one of the active ingredients of AS, has a clear structure and many physiological functions. EE appeared to have positive effects on the nervous system, such as enhancing memory in sleep deprivation mice[14], alleviating amnesia in isoflurane-induced cognitive dysfunction[15], and protective effects against neurotic atrophy[16]. EE has a protective effect on cognitive and memory damage induced by radiation. Whether EE can regulate the intestinal flora to protect nerves is still unknown.

In the present study, we hypothesize that this protective effect starts from affecting the intestinal microbiota and the brain-gut axis mechanism of EE to protect the cognitive and memory damage caused by radiation needs further study. EE was given to mice as a dietary supplement for 4 weeks, and then the mice were irradiated with a total dose of 4 Gy of $^{60}Co$-$\gamma$ rays. The relationship between gut microbiota and cognition of irradiated mice under EE supplementation was explored through a combination of behavioral tests, microecology, and metabolomics. Next, fecal microbiota transplantation (FMT) experiments were used to verify the mechanism of EE regulating the brain-gut axis.

## Results

### EE supplementation prevents radiation-induced cognitive impairment and damage of neuron morphology and ultra-structures in hippocampus.
To investigate whether dietary EE supplementation alleviates radiation-induced nerve damage, behavioral tests were conducted. The results of the water maze experiment showed that EE supplementation significantly reduced the time of arrival to the platform in irradiated mice, and decreased the mistakes as well ($p < 0.05$, Fig. 1b). In the open filed experiment, EE intake reduced dwell time in the center of irradiated mice ($p < 0.05$, Fig. 1b). In addition, EE supplementation alleviated the anhedonia caused by radiation compared to the model group, ($p < 0.05$, Fig. 1b). However, novel object recognition test (NORT) showed no significant change between model group and EE group.

Changes of hippocampal neurons are directly connected with learning and memory activities. Figure 1c showed lots of neurons of hippocampus arranged neatly in the control group. However,

radiation induced neuronal pyknotic. EE supplementation mitigated radiation induced neuronal damage (Fig. 1c). At the same time, normal nucleus and the mitochondria, Golgi bodies, and endoplasmic reticulum could be clearly observed by transmission electron microscopy (Fig. 1d). On the other hand, radiation caused neuronal nuclear membranes to be unclear, mitochondria to swell, and vacuoles to appear (Fig. 1d). The ultrastructural features of hippocampus in EE group changed slightly (Fig. 1d).

### EE intake contributes to maintaining intestinal barrier integrity in irradiated mice.
Following the EE supplementation alleviating neuron damage, we detected whether EE had any impact on the intestinal barrier. First, the HE staining of the colon showed that EE intake alleviated the pyknosis of epithelial cells, the inflammatory hyperplasia of lamina propria, and deep staining of the nucleus induced by radiation (Fig. 2a). Next, from the TEM, the swollen mitochondria, pyknosis of nucleus, reduced microvilli, and partially broken tight junctions were observed in the model group (Fig. 2b). EE supplementation prevented the tight junction broken. Compared to the model group, EE intake significantly increased the expressions of occluding, ZO-1, and claudin in the colon (Fig. 2c). Furthermore, we found the consumption of EE could decrease the levels of the inflammatory factor. Given the changes in the colon barrier and inflammatory factors, we hypothesized that the composition of gut microbiota would change accordingly.

### EE intake reshapes the gut microbiota in irradiated mice.
In view of the above studies, we found that EE can prevent the behavioral and histopathological damage of irradiated mice. We reasoned the composition of gut microbiota was responsible. Whether the supplementation of EE reshaped the gut microbiota composition of irradiated mice would be investigated in this section. Figure 3a directly showed the proportions of different taxonomic units in different groups. The proportion of Bacilli in EE group was higher than other groups. From the PCoA analysis (Principal coordinates analysis), these three groups showed a separation trend (Fig. 3b). The Figure showed the relative abundance of the intestinal microbiota at the different levels. The dominant phylum of the control group was Firmicutes, Bacteroidetes, Proteobacteria, and Cyanobacteria (Fig. 3c). However, in the model group, the relative abundance of Cyanobacteria decreased (Fig. 3c). In the EE group, the Proteobacteria also decreased (Fig. 3c). The top three relative abundance at the genus level were Lactobacillus, Helicobacter, and Bacteroides (Fig. 3f). EE supplementation prevented a radiation-induced decrease in the relative abundance of lactobacillus (Fig. 3f). LEfSe was carried out to identify the marker gut microbiota among the three groups. For the genus level, Helicobacter, Alistipes and Oscillospira were the maker bacteria in the model group. However, Lactobacillus, and Pediococcus were enriched in the EE group (LDA ≥ 3, Fig. 3h). It indicated that EE supplementation prevented the radiation-induced decrease of beneficial bacteria. As the PICRUSt2 showed the neurodegenerative diseases was included in the level two (Fig. 3i).

### EE supplementation changed the fecal metabolites in irradiated mice.
We conducted the untargeted metabolome analysis to screen the major different metabolites between EE group and model group using UHPLC-Q-TOF MS. According to the chemical taxonomy, the top 5 proportion of metabolites were organic acids and derivatives 17.927%, lipids, and lipid-like molecules 9.287%, organic oxygen compounds 7.991%, organoheterocyclic compounds 6.479%, nucleosides, nucleotides, and analogs

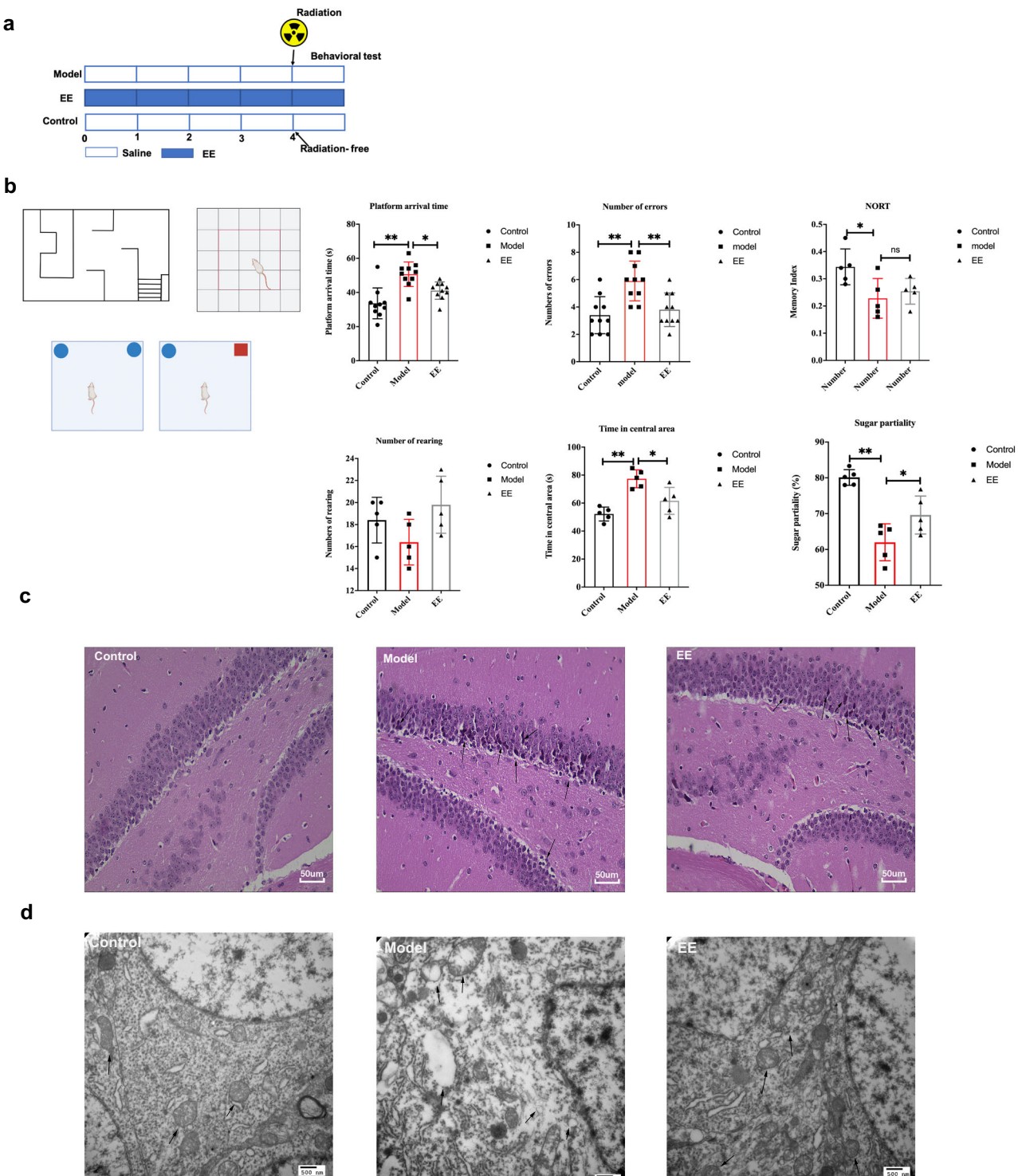

**Fig. 1 EE supplementation ameliorated radiation-induced cognitive impairment and damage of neuron in hippocampus. a** Design of the experiment. **b** Behavioral performance of the mice ($n = 5$ per group, *$p < 0.05$, **$p < 0.01$). **c** Histopathological analyses of hippocampus (H&E, scale bar $= 50\,\mu m$). **d** The ultrastructural features in hippocampus (TEM, scale bar $= 500\,nm$.). Values are expressed as the mean ± SD ($n = 5$). Statistical analyses were conducted using the one-way ANOVA followed by Tukey's post hoc test, *$p < 0.05$, **$p < 0.01$.

4.104%, (Fig. 4a). Based on univariate analysis, all metabolites detected in positive and negative ion mode were analyzed for differences. The differential metabolites with FC (Fold Change) >1.5 or FC < 0.6 and $p < 0.05$ were displayed in the volcanic map (Fig. 4b). PCA analysis showed that the EE group and the Model group could be separated from each other (Fig. 4c). Variable Importance (VIP) > 1, $p$ value < 0.05 was used to screen the

significantly different metabolites based on the OPLS-DA. Compared with the model group, there were 178 significantly changed metabolites, among which 117 metabolites were upregulated and 61 metabolites were downregulated (Fig. 4d). Next, bioinformatics analysis was performed on the differential metabolites (Fig. 4e). The results of the metabolic pathway enrichment analysis revealed the GABAergic synaptic pathway, which was

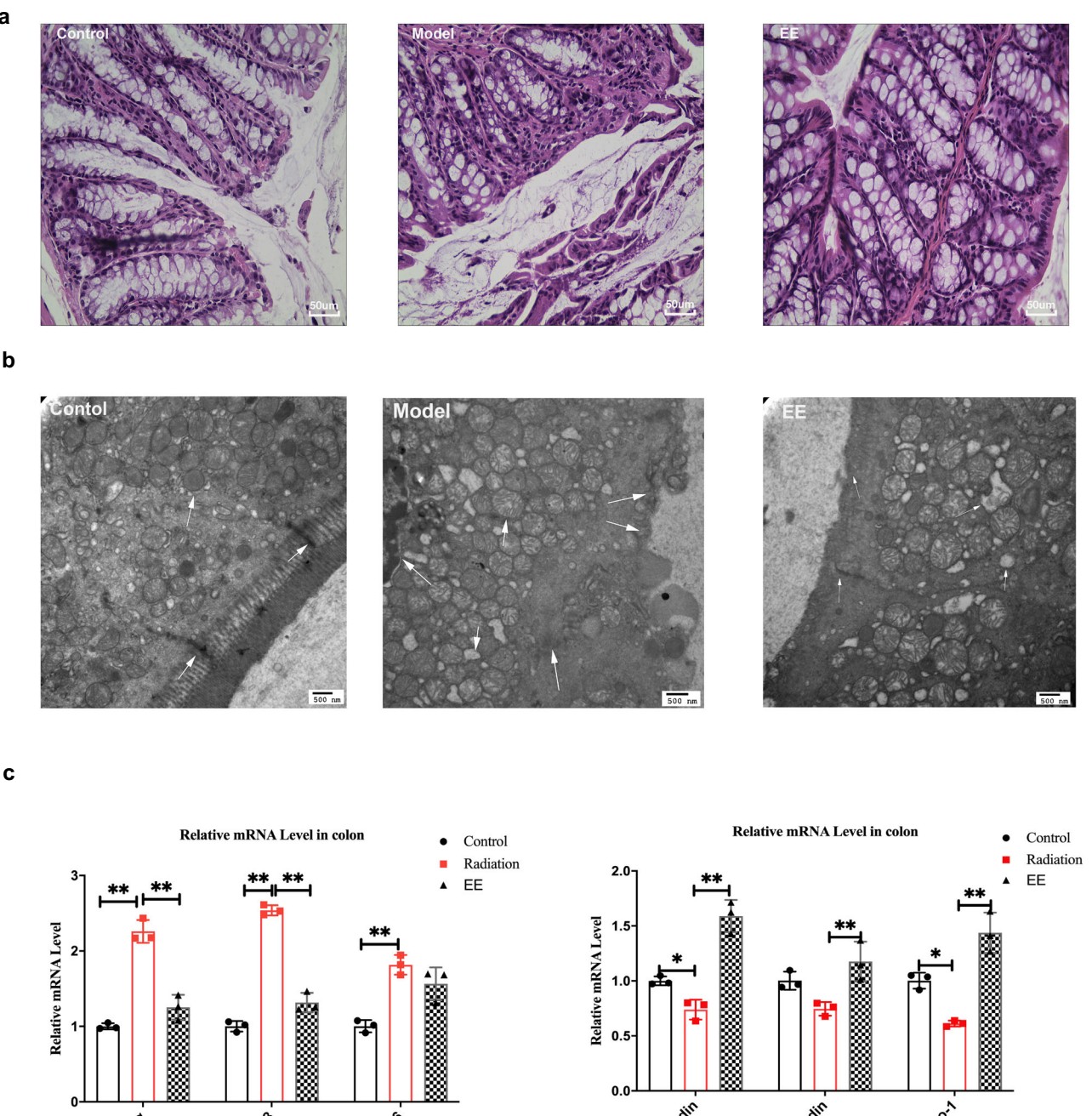

**Fig. 2 EE supplementation prevented colonic mucosa barrier impairment and inflammation in irradiated mice. a** The result of H&E staining on colon (Scale bar = 50um). **b** The ultrastructural features in colon (Scale bar = 500 nm). **c** The mRNA expression of inflammatory cytokines and tight junction proteins in colon of mice. (Values are expressed as the mean ± SD ($n$ = 5). Statistical analyses were conducted using the one-way ANOVA followed by Tukey's post hoc test, $*p < 0.05$, $**p < 0.01$).

related to the nervous system. Besides, it needs further verification of related pathways is needed.

In order to explore the relationship between the intestinal microbiota and its metabolites, we did correlation analysis on the metabolites and the gut microbiota with the relative abundance of top15. As the Fig. 4f showed, EE supplementation changed the levels of gut microbes and their metabolites in irradiated mice, and most of these changed metabolites belonged to organic acids and derivatives, which have a positive correlation with *Lactobacillus*, and negative correlation with *Helicobacter*, *Alistipes*, *Oscillospira*, *Streptococcus*, *Odoribacter*, *[Ruminococcus]* and *Aquabacterium*.

**EE intake alters neurotransmitter levels and their associated microbiome**. EE changed the levels of neurotransmitters related to learning and memory in hippocampus and colon of irradiated mice. The results showed that the behavioral changes of irradiated mice were accompanied by significant changes in neurotransmitters, among which, EE effectively prevented a decrease of 5-HT and ACH and an increase of the inhibitory neurotransmitter GABA caused by radiation (Fig. 5a). This was also mutually verified with the GABAergic synaptic pathway revealed by our metabolomics results. In addition, we also explored the associations between the intestinal microbiota and these changed neurotransmitters. The results (Fig. 5b) showed that 5-HT (5-hydroxytryptamine) was

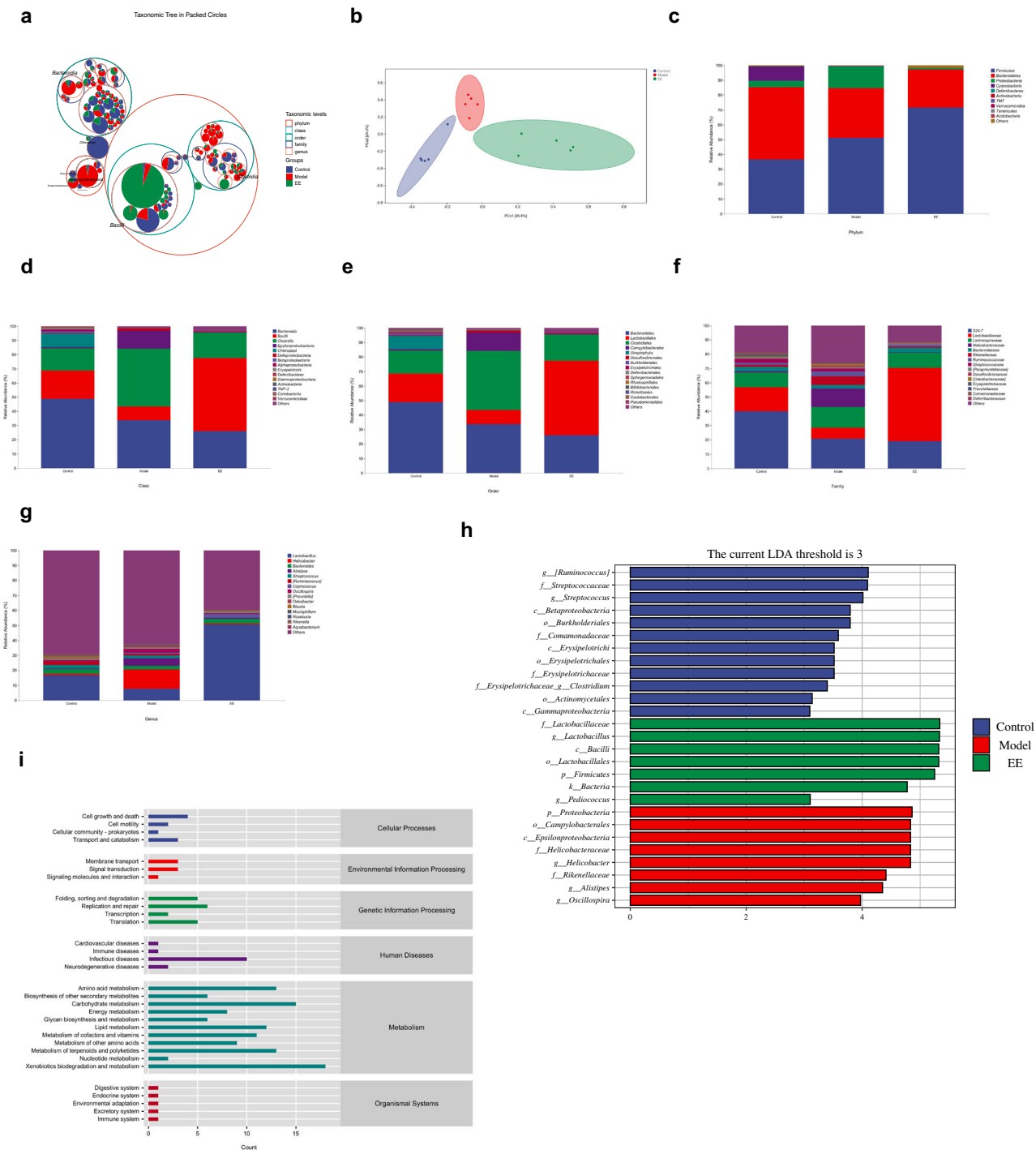

**Fig. 3 EE intake reshapes the gut microbiota in irradiated mice. a** Taxonomic tree in packed circles. **b** Principal coordinates analysis plot of Bray–Curtis. **c** The relative abundance of bacteria at phylum level; **d** the relative abundance of bacteria at class level; **e** the relative abundance of bacteria at order level; **f** the relative abundance of bacteria at family level. **g** The relative abundance of bacteria at genus level. **h** Linear discriminant analysis (LDA) effect size showing the most differentially significant abundant taxa enriched in microbiota among the EE group, model group, and control group. (LDA > 3). **i** Microbial functions were predicted by PICRUSt2 upon KEGG ($n = 5$).

negatively correlated with *Aquabaterium (p < 0.05)*. ACH was positively correlated with *Lactobacillus (p < 0.05)* and negatively correlated with *Oscillospira, Mucispirllum,* and *[Ruminoccus] (p < 0.05), Rikenella* and *Aquabaterium*. There is a positive correlation between GABA and *Mucispirllum (p < 0.05)*, and negative correlation between GABA and *Lactobacillus (p > 0.05)*. NE was

positively correlated with Lactobacillus (p < 0.05), and negatively correlated with Helicobacter and [Ruminoccus] (p < 0.05).

Therefore, we hypothesized that EE regulates the behavior and effect the congnition of irradiated mice via gut microbiota - neurotransmitter-brain communication. Next, we verified this hypothesis through fecal microbiota transplantation (FMT) experiment.

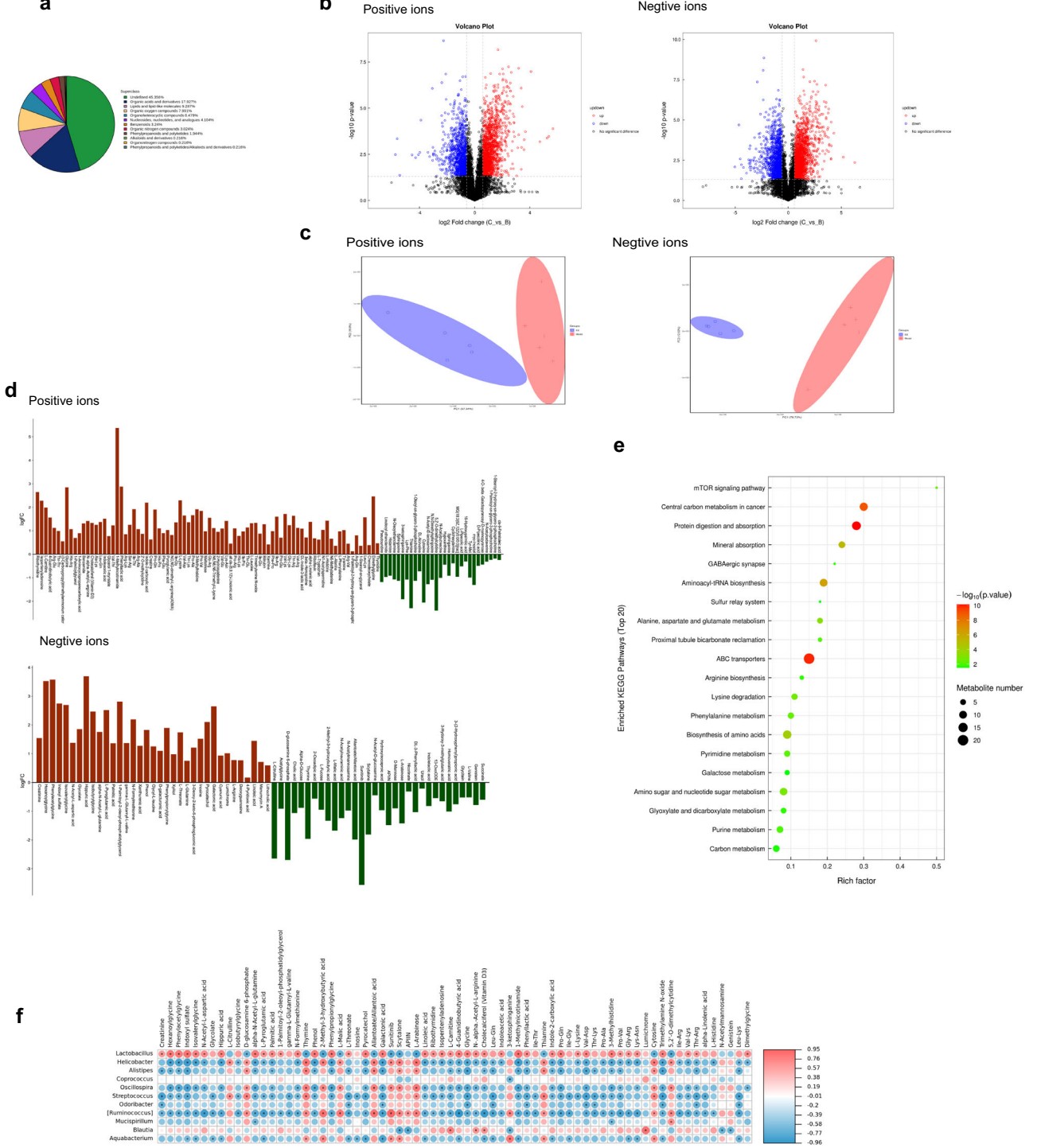

**Fig. 4 EE supplementation changed the fecal metabolites in irradiated mice. a** Statistics of chemical Taxonomy of metabolites. **b** The volcano plot of the fecal metabolites. (Metabolites with FC > 1.5 and *p* value < 0.05 were represented in red; metabolites with FC < 0.67 and *p* value < 0.05 were represented in blue. Non-significant metabolites are shown in black). **c** The PCA plot of the fecal metabolites. **d** Differential multiple analysis of metabolites with significant differences. (OPLS-DA, VIP > 1 and *p* value < 0.05, student's t-test). **e** The results of the metabolic pathway enrichment analysis. **f** Correlation analysis among the gut microbiota and metabolites with significant differences. Spearman's correlation analysis was performed by data of the Model group and EE group (*n* = 5).

**FMT from EE-fed mice regulated gut microbiota composition.** To determine whether we successfully transplanted the microbiota of EE donor into the mice, we analyzed the gut microbiota composition. As the results showed in Fig. 6, the proportion of *Bacilli* in FMT_EE group was similar to the EE group and both of them were higher than other groups (Fig. 6a). The dominant

phylum of FMT_EE group were *Firmicutes, Bacteroidetes* and *Proteobacteria*, which was similar to the EE-fed mice (Fig. 6b). At class level and order level, the dominant microbiota of FMT_EE group was consistent with that of EE group respectively (Fig. 6b). The top three relative abundance at the genus level was *lactobacillus, bacteroides*, and *helicobacter* (Fig. 6b).

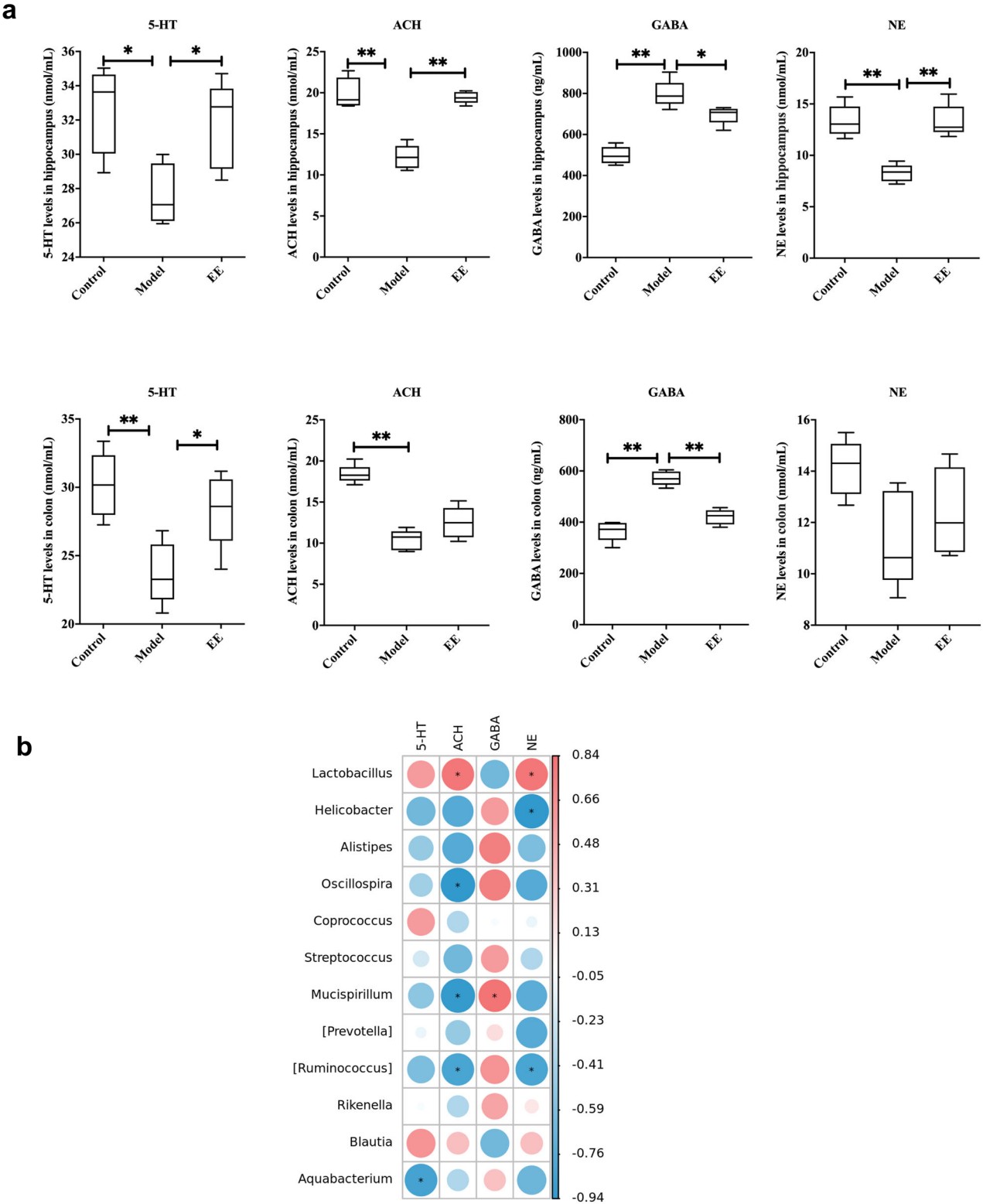

**Fig. 5 EE intake alters neurotransmitter levels and their associated microbiome. a** The levels of neurotransmitters in hippocampus and colon. Values are expressed as the mean ± SD ($n = 5$). Statistical analyses were conducted using the one-way ANOVA followed by Tukey's post hoc test, *$p < 0.05$, **$p < 0.01$. **b** Correlation analysis among neurotransmitters in hippocampus and gut microbiota. Spearman's correlation analysis was performed by data of the Model group and EE group.

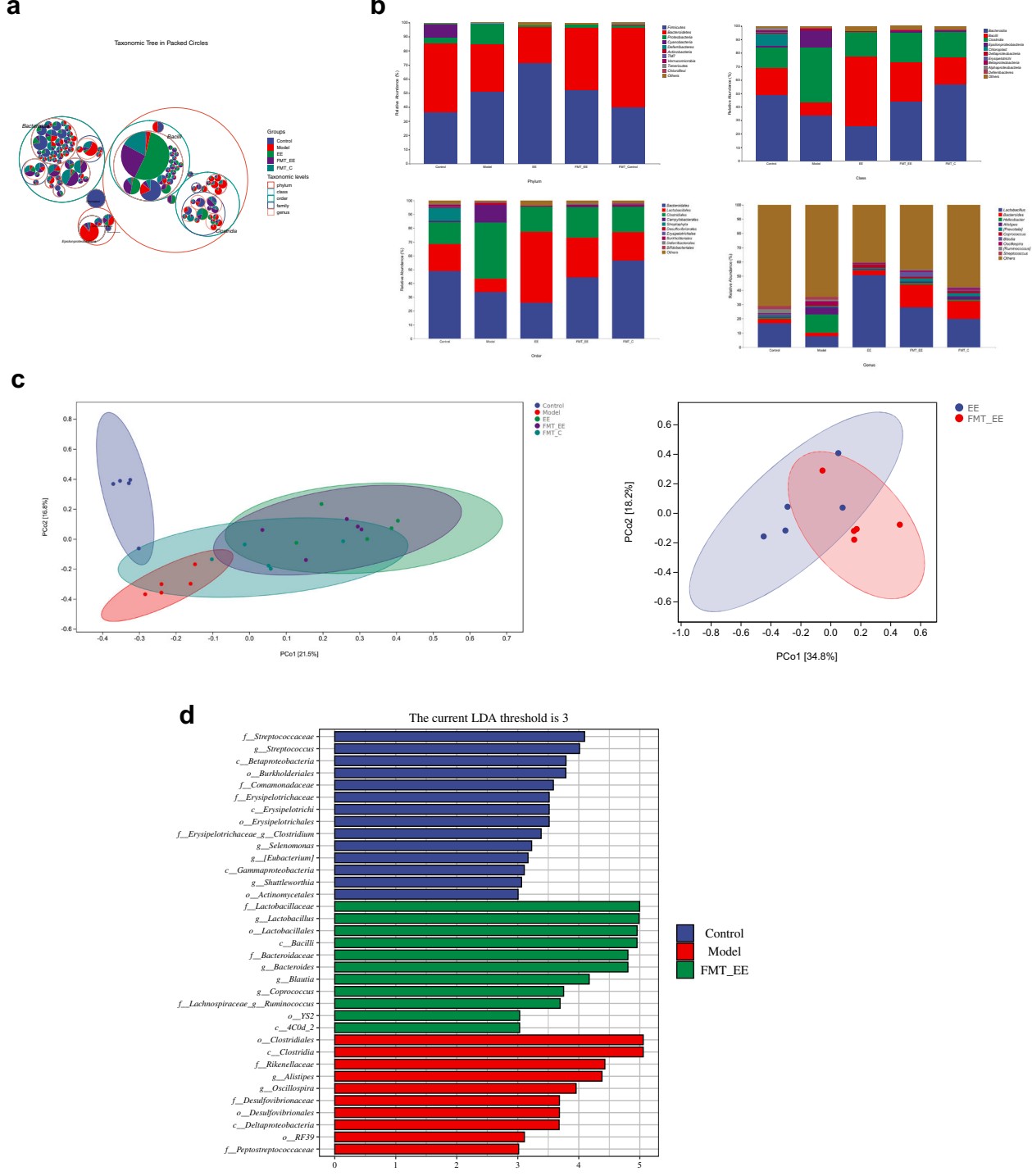

**Fig. 6 FMT from EE-fed mice regulated gut microbiota composition. a** Taxonomic tree in packed circles. **b** The relative abundance of bacteria at different level; **c** principal coordinates analysis plot of Bray–Curtis. **d** Linear discriminant analysis (LDA) effect size showing the most differentially significant abundant taxa enriched in microbiota among the model group, control group, and FMT_EE group (LDA > 3).

From the PCoA analysis, EE group overlapped with FMT_EE group, and was separated from model group and control group, respectively (Fig. 6c). It indicated that the fecal microbiome of the EE donor was successfully transferred into mice. The Lefse revealed that *Lactobacillus, Bacteroides, Ruminococcus, Blautia, and Coprococcus* were the marker microbiota at genus level (Fig. 6d).

As mentioned above, we also observed the effect of EE fecal transplantation on colon in mice damaged by radiation. FMT of

EE alleviated the colonic damage and enhance the tight junctions (Fig. 7a). We found that FMT of EE reduced the breakdown of tight junctions and increased the mRNA expression of occluding, claudin, and *Zo-1* ($p < 0.05$, Fig. 7b), while significantly inhibiting the expression of inflammatory cytokines (*TNF -α, IL-1β*, $p < 0.05$, Fig. 7b). However, compared to the model group, there was no significant differences in the expression of the tight junction proteins and inflammatory factors in FMT_control group (Fig. 7b).

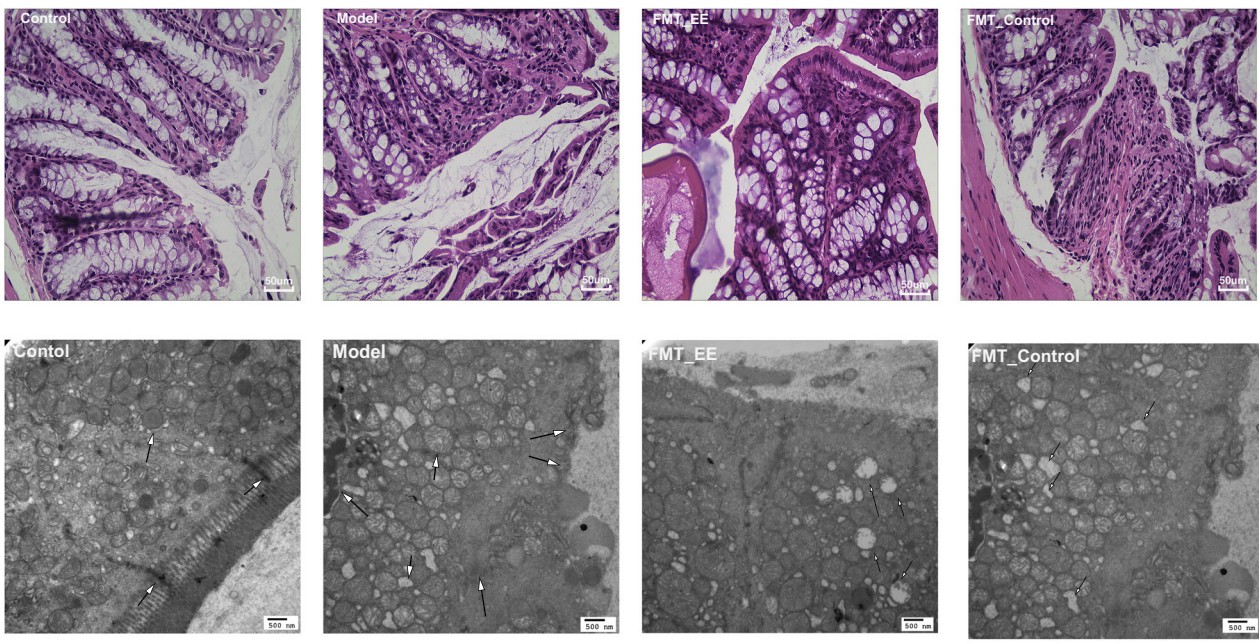

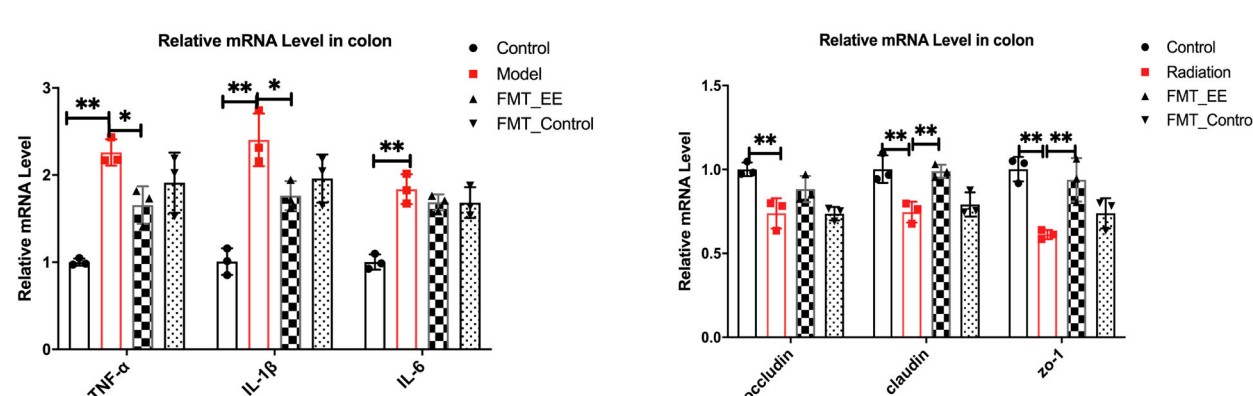

**Fig. 7 FMT from EE-fed mice prevented colonic mucosa barrier impairment and inflammation in irradiated mice. a** Histological features in colon. (H&E, scale bar = 50 μm, TEM, scale bar = 500 nm). **b** The mRNA expression of inflammatory cytokines and tight junction proteins in colon of mice. Values are expressed as the mean ± SD ($n = 5$). Statistical analyses were conducted using the one-way ANOVA followed by Tukey's post hoc test, *$p < 0.05$, **$p < 0.01$.

**FMT from EE-fed mice prevented cognitive impairment and damage of neuron morphology in irradiated mice.** We transplanted the feces of donor mice to the receipt mice for 4 weeks to see if the gut microbiome affected nerve injury, followed by the same behavior experiments. The results indicated that the mice of FMT _ EE group performed significantly better in the behavior test compared to the model group (Fig. 8b). Similarly, the HE results showed that FMT of EE decreased the pyknotic cells compared to the model group. From the TEM, we observed less mitochondrial swelling and clearer nuclear membrane in the FMT_EE group (Fig. 8c). However, there was no obvious morphological differences and behavioral changes between FMT_contol group and model group (Fig. 8b, c).

**FMT from EE-fed mice affected the neurotransmitters and their associated microbiome.** In view of the fact that EE intake affected neurotransmitters and microbiota, and the transplantation of microbiota alleviated radiation-induced cognitive impairment, we investigated the level of neurotransmitter in this part. There are no obvious changes of neurotransmitters between the FMT_control group and model group. FMT_EE significantly increased the level of ACH, GABA, and NE in hippocampus ($p < 0.05$, Fig. 9a). What's more, 5-HT was significantly positively correlative with *lactobacillus* ($|R| > 0.829$, $p < 0.05$, Fig. 9b). ACH was positively relative with *Bacteroides* and *Parabacteroides* ($|R| > 0.829$, $p < 0.05$), and negatively correlative with *Alistipes* and *Mucispirllum* ($|R| > 0.886$, $p < 0.05$, Fig. 9b). GABA was negatively correlated with *Bacteroides* and *Parabacteroides*, and positively correlated with *Alispes* and *Mucispirillum* ($|R| > 0.886$, $p < 0.05$, Fig. 9b). NE was positively correlated with *Blautia* ($|R| > 0.826$, $p < 0.05$, Fig. 9b).

**EE intake regulated the PKA/CREB/BDNF signaling through gut microbiota.** Based on the above microbiota, metabolic and behavioral analysis, we next explored whether EE could affect PKA/

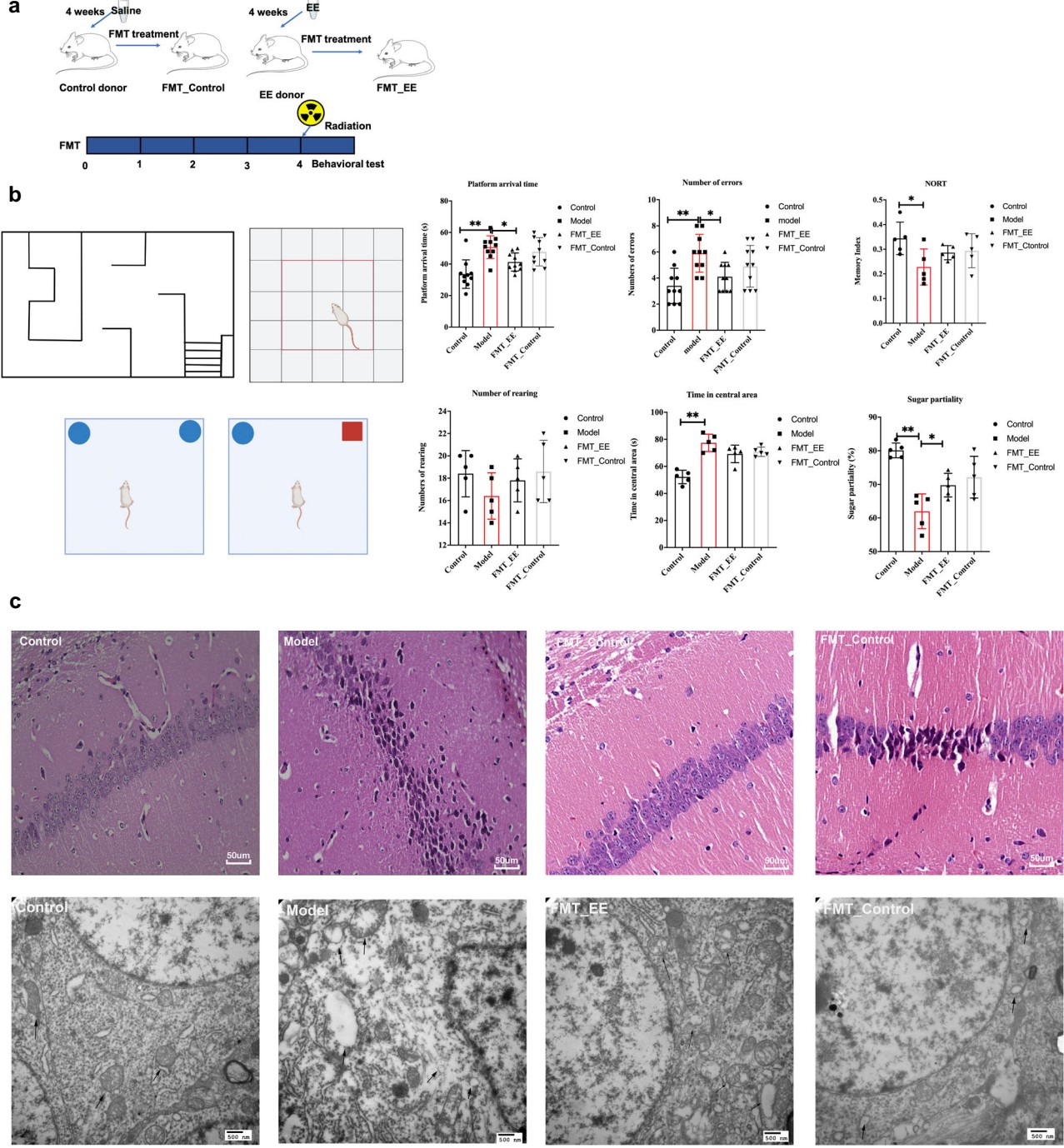

**Fig. 8 FMT from EE-fed mice prevented cognitive impairment and damage of neuron morphology in irradiated mice. a** Schematic of experimental design. **b** Behavioral performance of FMT mice. (Values are expressed as the mean ± SD ($n = 5$). Statistical analyses were conducted using the one-way ANOVA followed by Tukey's post hoc test, *$p < 0.05$, **$p < 0.01$). **c** Histopathological features of hippocampus of FMT mice (H&E, scale bar = 50 μm, TEM, scale bar = 500 nm).

CREB/BDNF signaling through intestinal microbiota. The total protein was obtained from the hippocampus. The relative protein levels were detected by western blot. No obvious changes in the protein level of PKA in this study (Fig. 9c). The relative protein level of BDNF showed a decrease in the model group. EE intake and FMT_EE significantly increased the relative protein level of BDNF in irradiated mice. Meanwhile, the CREB level change little in EE and FMT_EE groups compared with the model group. But the CREB phosphorylation expression was increased in these two groups (EE group $p < 0.05$, and FMT_EE $p < 0.01$, Fig. 9c).

## Discussion

Studies have reported that radiation has a serious damage on both gut and cognition. The gut microbiota was viewed as a novel therapeutic target for brain disorders and as critical windows in neurocognitive development[17–19]. Our results showed that EE could effectively prevent cognitive decline in irradiated mice (Fig. 1). We hypothesize that the cognitive decline and neurological damage is due to changes in gut microbiota. EE can prevent cognitive impairment and neurological damage by reshaping the intestinal microbiota. In our study, the EE intake remodeled the

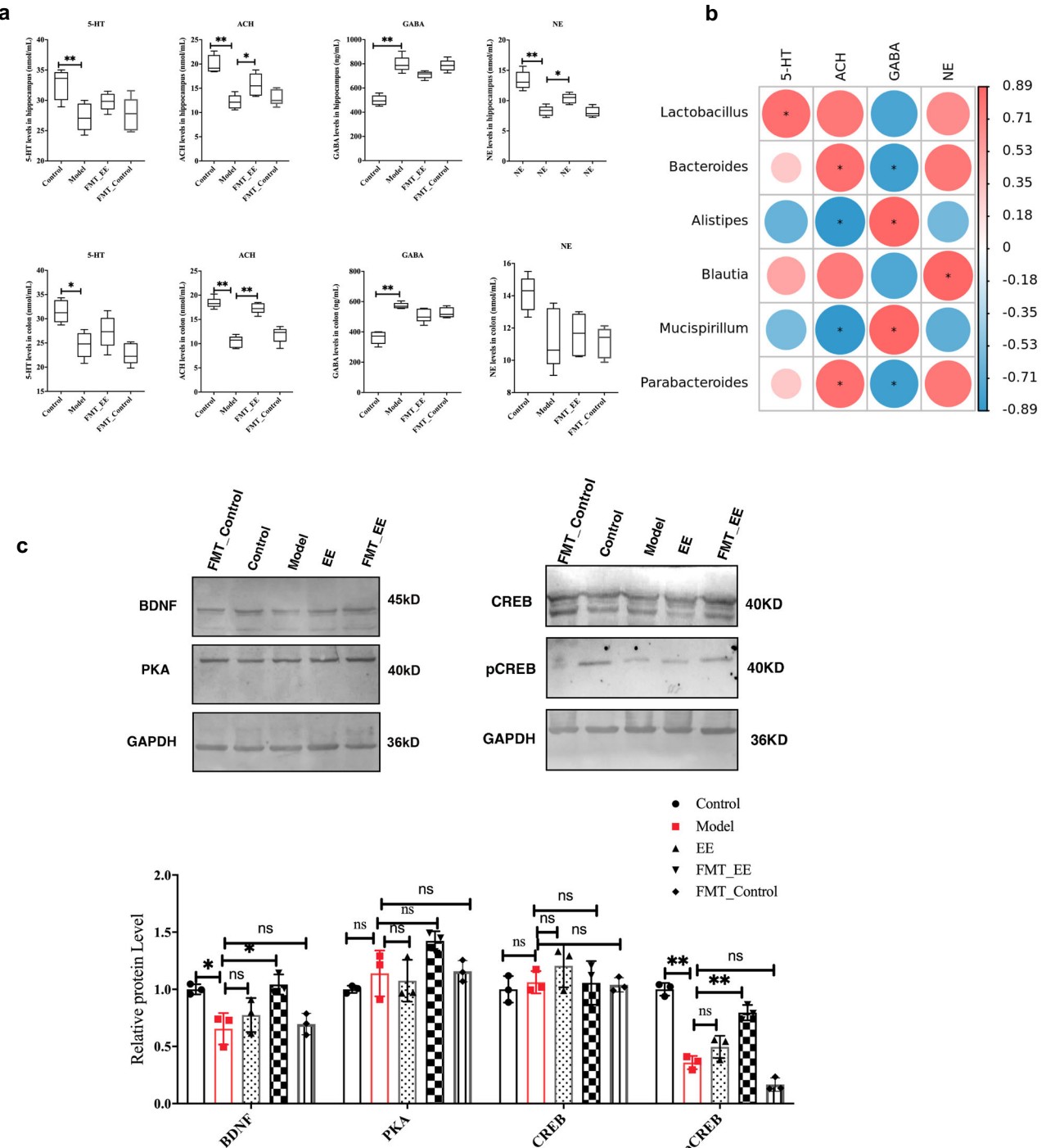

**Fig. 9 FMT from EE-fed mice affected the neurotransmitters and their associated microbiome and regulated the PKA/CREB/BDNF signaling. a** The levels of neurotransmitters in hippocampus and colon in FMT mice. (Values are expressed as the mean ± SD ($n = 5$). Statistical analyses were conducted using the one-way ANOVA followed by Tukey's post hoc test, $*p < 0.05$, $**p < 0.01$). **b** Spearman's correlation analysis among neurotransmitters and gut microbiota was performed by data of the Model group and FMT_EE group. **c** The relative protein levels in different groups. (Values are expressed as the mean ± SD ($n = 3$). Statistical analyses were conducted using the one-way ANOVA followed by Tukey's post hoc test, $*p < 0.05$, $**p < 0.01$; the samples derive from the same experiment or parallel experiments and that blot was processed in parallel.).

microbiota of irradiated mice (Fig. 3a–f). Altered intestinal permeability and disruption of the mucosal barrier can lead to an imbalance in the intestinal flora, while the intestinal epithelium and intestinal microbiota interact directly with the intestinal immune system[20–22]. We have discussed the reasons for the changes in the gut microbiota from many aspects. First, the pathological results showed that the supplementation of EE

prevents intestinal damage caused by radiation, including preventing the destruction of tight junctions. Secondly, at the genetic level, it was detected that the level of tight junction protein obviously increased in the EE group (Fig. 7). It is precisely because EE protects the integrity of the gut that it avoids dysregulated intestinal microbiota caused by radiation and the decrease of pro-inflammatory factors. In addition, EE supplementation

increased the relative abundance of *Lactobacillus*, which has been demonstrated to attribute to improve cognition[23–25]. Indeed, *Lactobacillus* was emphasized in the transporter-based mechanisms of 5-HT. Both supplementary EE and FMT of EE increased the relative abundance of *Lactobacillus* (Figs. 3h and 6d). On the other hand, the colonization with *Helicobacter* can often remain harmless and has been implicated in the development of various cognitive-related diseases. For example, an increase of helicobacter can be observed in chronic psychosocial stress, *Helicobacter pylori* infection enhance the development of PD symptoms, and the relative abundance of Helicobacter increased in AD mice[8]. *Alistipes* was also the marker microbiota in model group, which was found to be an increasing abundance in the depression. Our results showed a decrease of *Helicobacter and Alistipes* in EE supplementation and FMT of EE (Figs. 3g, 6d), which revealed that EE reduced the risk of cognitive impairment by regulating the intestinal microbiota. Since several neurotransmitters were involved in cognition and behavior such as GABA, ACH, and others have been demonstrated to synthesize by microbiota[8,26,27], we found that the changes in these neurochemical signals do have a certain correlation with microbiota (Fig. 5b). Furthermore, later fecal transplant experiments also verified this result (Fig. 9a). *Ruminococcus* was involved in the synthesis of tryptamine, which induced the 5-HT secretion[28]. We found it was the marker bacteria in FMT_EE group. As studies have reported that *Streptococcus* was related to GABA synthesis[29,30], in our study, correlation analysis showed that *Streptococcus* was related to GABA levels (Fig. 5b). Neurotransmitters were responsible for the main signal transmission between types of neurons and glial cells to control the functional brain activity[31]. Unlike the neurotransmitters, their precursors involved in neurotransmitter synthesis have been reported to cross the blood-brain barrier, and these substances can be derived from bacteria encoding specific enzymes to catalyze the conversion of substrates[32–34]. As we found in the metabolite analysis of the microbiota, EE intake increased the level of L-Tryptophan and L-Phenylalanine (Fig. 4d) which is the precursors of 5-HT and Phenylethylamine respectively and promote the intestinal motility[35]. Many researchers have demonstrated that neurotransmitters such as GABA, 5-HT, NE, and ACH are all related to cognitive impairment[36,37] and *Bacteroides*, *Lactobacillus* and *Ruminococcus* were changed significantly in patients with neurodegenerative diseases[38,39]. The results of the metabolic pathway enrichment analysis revealed the GABAergic synaptic pathway (Fig. 4). In the next experiment, we also verified that GABA (Fig. 5) and glutamine (Fig. 4) along with their related microbiota have also changed. Brain-derived neurotrophic factor (BDNF) has a prominent role in nervous system, including regulating learning and memory ability and neurotroplasticity[40,41]. Similarly, studies have demonstrated that gut microbes not only affect neurotransmitter synthesis in the brain but also regulate the expression of BDNF in the brain. Moreover, BDNF can regulate synaptic transmission and induce presynaptic release of glutamate and GABA[42,43]. It may be a possible mechanism for the microbiome-gut-brain axis. PKA/CREB/BDNF signaling plays an important regulatory role in neurodevelopment, cognition and emotion[44]. Therefore, we assumed whether EE affected this pathway through microbiota. FMT experiment was used to verify a more robust inference that EE supplementation regulated PKA/CREB/BDNF pathway via the gut microbiota. In our present study, donor mice were given EE for 4 weeks, and then their feces were transplanted into recipient mice for 4 weeks to see if FMT of EE also affected this pathway. We observed a decreased expression of BDNF and CREB phosphorylation in model group. However, EE intake and FMT of EE activated the PKA/CREB/BDNF signaling along with the alteration of gut microbiota. Therefore, regulating the

microbiota to activate the PKA/CREB/BDNF signaling could be a promising therapeutic strategy for the prevention of radiation-induced cognitive damage.

In summary, the current study shows a link between gut microbiota and cognitive function via microbial metabolites especially neurotransmitters and PKA/CREB/BDNF signaling under the supplementation of EE in irradiated mice. EE intake protected the tight junction of the colon and remodeled the gut microbiota, especially the change of *lactobacillus*, altered the microbial metabolites including precursors of neurotransmitters, protected the ultrastructure of hippocampus. Furthermore, the fecal transplantation of EE donors verified that EE alleviated cognition and spatial memory impairments, and activated the PKA/CREB/BDNF signaling via the gut microbiota. These findings evidence the potential therapeutic value of EE in radiation-induced cognitive and memory impairments and emphasize the novel mechanism of EE effect on the gut-brain axis.

## Methods

**Study design**. The aim of this study was to elucidate the mechanism of gut-brain axis by which EE acts on prevention of radiation-induced cognitive and memory damage. Adult male Kunming mice were purchased from the Animal Experimental Center of the 2nd Affiliated Hospital of Harbin Medical University (Harbin, Heilongjiang, China, the certification number: SYXK (HEI) 2019-001) and housed in an acclimated breeding room at $22 \pm 2\,°C$ with a regular 12 h light cycle. All of the experimental animal procedures were evaluated and approved by the local ethics committee of Harbin Institute of Technology (IACUC-2021075). After adaptive feeding for one week, the animals were randomly grouped: control group, model group, and EE group. There are 10 mice in each group. The mice of EE group were administrated with EE ($C_{34}H_{46}O_{18}$, Beijing Solarbio Science& Technology Co., Ltd, the purity was 98% detected by HPLC) at the dose of 50 mg/kg/d for 4 weeks. The control group and model group were administrated with saline instead. And then, the mice of radiation group and EE group were irradiated by $^{60}Co\text{-}\gamma$ ray at a total dose of 4 Gy (0.9 Gy/min). The control group received no radiation treatment.

The fecal transplant was performed based on an established protocol[45]. Briefly, the donor mice of EE were administrated with EE for 4 weeks. And the control donors were administrated with saline for 4 weeks. Collection the feces of the mice daily and resuspended 100 mg in 1 ml of sterile saline, vortex for 10 s, centrifuged at $800 \times g$ for 3 min, and finally collected the supernatant as a transplant material. To prevent the bacterial composition changes, the feces were prepared daily and gavage to mice within 10 min. The FMT mice were gavaged with the fresh transplant material daily for 4 weeks. After four weeks fecal transplantation, the mice were irradiated by $^{60}Co\text{-}\gamma$ ray at a total dose of 4 Gy.

## Behavioral test

*Water maze*. Water maze was used to evaluate the learning and memory ability according to our previous method[46]. Briefly, as shown in Fig. 1b, there are many blind zones in the rectangular water maze. In the experiment, the time of arrival time to the platform in exit and the error times of mice entering the blind zone were recorded. Five days of training followed by formal experiments.

*Novel object recognition*. The NORT was performed as previously reported with a little modified[47]. The experiment was carried out in a box ($45 \times 30 \times 20$ cm, Fig. 1b). The test was divided into habituation period, familiarization section, and test section. The first and second days were the habituation period: each mouse was placed individually in the empty box for 5 min. Days 3–4 is the familiarization period: put two identical objects in a symmetrical corner of the box. The mice were put into the box and their exploration time for the new and old two different objects was recorded within 5 min. The fifth day was the test period: one of the objects was replaced with a new object with the same position. The mice were put into the box and their exploration time for the new and old two different objects was recorded within 5 min. The discrimination index $= (T_n - T_0)/(T_n + T_0)$ The time exploring the new object is represented by $T_n$, whereas $T_0$ represents the time exploring the old object.

*Open field test*. For open field test, mice were placed individually in a corner of an open-field box ($50 \times 50 \times 30$ cm) with black vertical walls and the bottom was equally divided into 25 grids. The top of the box is open and illuminated by natural light. The mice were placed in the central grid of the open field experiment box, and the vertical movement times of the rats and the activity time in the central area were observed and recorded.

*Sucrose preference test*. A sucrose preference test was used to assess the sensitivity of mice's nervous systems. The mice were fed two bottles of sugar water for 24 h

before the experiment, then one bottle of sucrose water was replaced with distilled water for the next 24 h. The mice were then given distilled water and sugar water for four hours after being denied water for 24 h. Sugar water partiality was calculated as the percentage of sugar water consumption to the sum of sugar and pure water consumption.

**Histologic morphology**. Hippocampus and colon tissue were fixed in 4% paraformaldehyde, and then dehydrated in ethanol and clarified with xylene. After embedded in paraffin, samples were sectioned at 4 μm thickness and stained with Hematoxylin & Eosin (HE). All the sections were observed and photographed under a light microscope (Eclipse Ci-L, Nikon,Japan).

**Transmission electron microscopy**. The tissues were sliced into 1 mm³ and fixed in 2.5% glutaraldehyde, postfixed with 1% osmic acid for 2 h, dehydrated in alcohol. Then, the samples were embedded in resin. The lead citrate was used to stain the ultrathin sections. The ultrastructures were viewed on a transmission electron microscope (H-7650; HITACHI; Tokyo, Japan).

**Neurotransmitters levels**. The level of neurotransmitters including serotonin (5-HT), norepinephrine (NE), γ-aminobutyric acid (GABA), and acetylcholine (ACH) were determined by ELISA kits (Shanghai Lengton Bioscience Co., LTD, Shanghai, China) according to the protocol.

**Real-time qPCR analysis**. A commercially RNA extraction kit (TaKaRa Universal RNA Extraction Kit, Dalian, China) was used to isolate the total RNA from hippocampus and colon tissues according to the manufacturer's instruction. The synthesis of cDNA uses PrimeScript™ RT reagent Kit (TaKaRa; Dalian, China) according to the manufacturer's instructions. The gene-specific primers used in the test were supplied in Supplementary Table 1. Reactions were performed using TB Green ® Premix Ex Taq ™ II (Tli RNaseH Plus) (TaKaRa; DaLian, China) on ABI QuanrStudio5 Real-time PCR Detection System (Life technologies, Singapore) with the following conditions: 95 °C for 30 s, followed by 40 cycles of 95 °C for 5 s, 60 °C for 30 s. The expressions were calculated as a relative value after normalization to β-actin mRNA.

**Composition of gut microbiota**. Total genomic DNA was isolated from the feces using the OMEGA Soil DNA Kit (D5625-01) (Omega Bio-Tek, Norcross, GA, USA), as directed by the manufacturer. The bacterial 16S rRNA gene's hyper-variable region V3–V4 was amplified using primer pairs F: ACTCCTACGG-GAGGCAGCA, R: CGGACTACHVGGGTWTCTAAT. Following the individual quantification phase, amplicons were pooled in equal proportions for pair-end 2 × 250 bp sequencing on the Illlumina MiSeq platform using the MiSeq Reagent Kit v3. The raw reads were uploaded to the NCBI Sequence Read Archive (SRA) database (accession number, PRJNA746423).

**Fecal metabolomics**. Before euthanasia, each mouse's feces were collected and quickly frozen using liquid nitrogen. Fecal samples and extraction reagent were added in the ratio of 80 mg:1 mL to extract metabolites. The samples were vacuum dried and re-dissolved in 100 μL acetonitrile/water (1:1, v/v) solvent and transferred to LC vials for LC-MS analysis.

**Western blot**. Total protein was isolated from the brain tissues and adjusted to the same final concentration using the protein extraction kit (Beyotime Institute of Biotechnology, Shanghai, China) and BCA Assay Kit (BCA Protein Assay Kit, Solarbio). SDS-PAGE was used to separate the samples, which were then transferred to NC (nitrocellulose filter) membranes. After being blocked with 2% BSA for 60 min, the NC membranes were incubated overnight with primary antibodies at 4 °C. The following day, after washing three times with PBST(phosphate buffered saline tween-20), the membranes were incubated with the secondary antibody (Goat Anti-Mouse IgG(H+L), Alkaline Phosphatase conjugate, and Goat Anti-Rabbit IgG(H+L), Alkaline Phosphatase-conjugated, Proteintech, China) for 1 h at room temperature. The membranes were washed four times in PBST buffer and incubated with a commercial western immunoblotting detection reagent (western blue stabilized substrate for alkaline phosphatase, Promega, USA) for 5 min. The resulting immuno-reactive bands were exposed by Amersham Imager 600 (GE). The primary antibodies used in this study were PKA (Abcam, ab108385,1:1000), CREB (Abcam, ab32515, 1:1000), pCREB (Abcam, ab32096, 1:1000), BDNF (Abcam, ab108319,1:1500), and GAPDH (Proteintech, 60004-1-Ig, 1:2000).

**Statistics and reproducibility**. All data were reported as means ± standard deviation (S.D) and analyzed using GraphPad Prism version 8.0 with one-way ANOVA (Analysis of Variance) and multiple-comparison test (Tukey's post hoc test); and significance was marked as *$p < 0.05$, **$p < 0.01$. We assessed correlations using Spearman's correlation coefficient.

QIIME2 and R packages (v3.2.0) were used in the microbiome analysis. To identify the marker bacteria in gut, we used LEfSe analysis and the LDA value ≥ 3. The variable importance in the projection (VIP) value >1 in the orthogonal partial least-squares discriminant analysis (OPLS-DA) and $p$ values < 0.05 (Student's t-test) were viewed as significantly different for fecal metabolomics.

**Reporting summary**. Further information on research design is available in the Nature Research Reporting Summary linked to this article.

## Data availability
The raw 16SrRNA sequence reads for fecal microbiota were deposited in the NCBI Sequence Read Archive (SRA) database (accession number, PRJNA746423). All the data are available in this paper. Source data of the graphs presented in the main figures and unprocessed blots are available in Supplementary Data 1. The datasets analyzed during this study are available from the corresponding author upon reasonable request.

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

## Acknowledgements

This research was supported by the National Key Research and Development Program of China: 2017YFC1601900 and Heilongjiang Touyan Team: HITTY-20190034.

## Author contributions

C.S. performed experiments, analyzed data, and prepared the manuscript. F.D. and Q.Y. assisted with experiments and data collection. T.J., D.Z., S.s., and Y.S. assisted with data analysis. Y.Z. oversaw the project and proofread the manuscript. W.L. designed the project, oversaw the experiments, and revised the manuscript. All authors read and approved the final manuscript.

## Competing interests

The authors declare no competing interests.
