## [Peer Review File · Communications Biology]

Reviewers' comments:

Reviewer #1 (Remarks to the Author):

In this study, the effect of Eleutheroside E on the cognition and gut microbiota of the 60Co-γray irradiated mice was evaluated by combining behavioral test, microecology and metabolomics. And the fecal microbiota transplantation (FMT) experiments are used to verify the EE activated the PKA/CREB/BDNF signaling via gut microbiota. The study is interesting and meaningful. But there are some issues need to be improved.

Minor comments:

1. Line 65, the gender of the mice should be indicated in the experimental design, and the grouping information should be detailed, such as the number of mice in each group.
2. The scale bar was unclear in Fig.1C, the same as Fig.2, Fig.6, Fig.7.
3. For the radiation dose, please explain why 4Gy was chosen.
4. Regarding the selected compounds, what was the basis for the selection?
5. In the discussion section, the changes in the microbiota of the model group and the EE group should be discussed more intensively, for example, the relevant functional roles of the marker microbiota need to be deeply explored.
6. It is not indicated whether bacteria that correlate with specific neurotransmitters are capable of synthesizing the neurotransmitters.
7. Check English grammar and spelling problems.

Reviewer #2 (Remarks to the Author):

Therapeutic doses of radiation are often associated with a detrimental impact on brain function, often manifesting as cognitive dysfunction in a number of domains. Eleutheroside E (EE), an important active ingredient from *Acanthopanax senticosus*, is associated with protective behavioral and biochemical effects in a number of model systems following various insults. Although EE has anti-inflammatory properties, the mechanisms underpinning these observations remains unclear. In this study, the authors test the hypothesis that a four-week EE supplementation is protective against irradiation induced cognitive dysfunction by remodelling the gut microbiome. This is comprehensively assessed using a variety of approaches including fecal microbiota transplantation experiments. The main findings reported are that the potential therapeutic value of EE in radiation induced cognitive and memory impairments is regulated by the gut microbiota via neurotransmitter systems and PKA/CREB/BDNF signalling pathways.

This is an interesting study with a number of potentially important observations. I have a number of queries, suggestions and recommendations:

- (1) From a conceptual perspective, the authors need to be much clearer about what hypothesis they are testing here. The introduction mentions mobile phone and space radiation for which is a pretty spurious association when they are really evaluating therapeutic doses of radiation. This needs to be explained in a much clearer and explicit way in the introduction.
- (2) The details supplied about the fecal microbiota transplant are very sparse and it is difficult to fully interpret until a more detailed account is provided. I would direct the authors to the following publication: Guidelines for reporting on animal fecal transplantation (GRAFT) studies: recommendations from a systematic review of murine transplantation protocols (<https://www.tandfonline.com/doi/full/10.1080/19490976.2021.1979878>). The information recommended in the checklist provided here should be included in a revised submission. In

particular, it is important to know what steps, if any, were taken to preserve the viability of the microorganisms in the sample. It also appears as though the authors did not deplete the resident microbiota prior to the initiation of the microbiota transplant. Of further concern is that a critical control group is missing since if I understand correctly, it is only the EE treated group that act as donors. Without this group, it is difficult to know if there is a difference in behavioural and molecular profiles between the control and EE-remodelled microbiota.

(3) Appropriate referencing is an issue throughout the manuscript. For example, authors cite a review paper about clinical FMT as the origin of the 'established protocol' they use in their experiments. The papers cited in the introduction regarding radiation all appear to be in the context of therapeutic doses even though they are linked to mobile phone radiation.

(4) The methods section should provide the approval number of the ethics approval.

(5) The experimental details for the behavioural assays are also lacking in detail. What lighting conditions were used for the open field. Please also see this paper for the experimental parameters that should be reported for the novel object recognition (Object recognition test in mice <https://www.nature.com/articles/nprot.2013.155>).

(6) The authors have used β -actin as their housekeeper gene – did they verify that its expression was not altered by the experimental interventions?

(7) I'm not clear on the relevance or value of the correlations between ACH and GABA levels in brain and colon tissue. I would recommend removing this from a revised manuscript.

Dear editor and reviewers:

Thank you very much for your constructive suggestions with regard to our paper " **Eleutheroside E supplementation prevents radiation-induced cognitive impairment and activates the PKA/CREB/BDNF signaling via gut microbiota** " (COMMSBIO-22-0044). We express our sincere gratitude to the editors' and reviewers' conscientious work in the whole process, your comments and/or suggestions are so important for the improvement of our manuscript. We have tried our best to revise and improve the manuscript according to the reviewers' good comments. The contents related to the reviewers' comments and revisions are marked in red throughout the revised manuscript. Responses to the reviewers' comments are as followed:

Reviewers' comments:

Reviewer #1:

In this study, the effect of Eleutheroside E on the cognition and gut microbiota of the 60Co-γray irradiated mice was evaluated by combining behavioral test, microecology and metabolomics. And the fecal microbiota transplantation (FMT) experiments are used to verify the EE activated the PKA/CREB/BDNF signaling via gut microbiota. The study is interesting and meaningful. But there are some issues need to be improved.

Minor comments:

1. Line 65, the gender of the mice should be indicated in the experimental design, and the grouping information should be detailed, such as the number of mice in each group.

Thank you very much for your valuable suggestions. Adult male KM mice were obtained from the Animal Experimental Center of the 2nd Affiliated Hospital of Harbin Medical University (Harbin, Heilongjiang, China); the certification number was SYXK (HEI) 2019-001. All of the experimental animal procedures were evaluated and approved by the local ethics committee of Harbin Institute of Technology (IACUC-2021075). All animals were maintained in an environmentally controlled breeding room with a regular 12-h light cycle at 22 ± 2 °C. After a normal diet for one week, the animals were divided into three groups: control group, model group, and EE group. There are 10 mice in each group.

2. The scale bar was unclear in Fig.1C, the same as Fig.2, Fig.7, Fig.8.

Thank you very much for your valuable suggestions. I have modified it in the revised manuscript.

Figure1

Figure 2

Figure 7

Figure 8

3. For the radiation dose, please explain why 4Gy was chosen.

Answer:

Thank you very much for your suggestions. Because of the uncertainty in extrapolating dose thresholds from rodents to humans, we should find a proper dose that impact CNS significantly according to the previous study. In addition, our laboratory had conducted experiments to investigate the effects of ⁶⁰Co γ -rays on brain injury at different dose rates and total doses ^{1,2}. The results showed that the biological effects of space radiation can be simulated at the dose rate of 0.9 Gy/min and total dose of 4 Gy. In our study, we used the total dose of 4 Gy, which caused not only brain damage but also gut microbiota disturbance. Actually, we just simulated the biological effects of space radiation instead of space radiation dose.

4. Regarding the selected compounds, what was the basis for the selection?

Answer:

Thank you very much for your valuable suggestions. Our laboratory conducted preliminary research on the protective effect of *Acanthopanax senticosus* functional components on irradiated mice, and found that Eleutheroside E has a positive effect, so this experiment selected EE for research.

5. In the discussion section, the changes in the microbiota of the model group and the EE group should be discussed more intensively, for example, the relevant functional roles of the marker microbiota need to be deeply explored.

Answer:

Thank you very much for your valuable suggestions. We have re-written this part according to your suggestion.

In addition, EE supplementation increased the relative abundance of *Lactobacillus*, which has been shown to play an important role in cognition ^{3,4,5}. Indeed, transporter-based mechanisms for the uptake of extracellular 5-HT were recently highlighted in a probiotic strain of *Lactobacillus*. Both supplementary EE and FMT of EE were increased the relative abundance of *Lactobacillus* (Fig.3H and Fig.6D). In the other hand, the colonization with *Helicobacter* can often remain harmless, and has been implicated in the development of various cognitive-related diseases. For example, chronic psychosocial stress induced an increase of *Helicobacter* ⁶, *Helicobacter pylori* infection enhance the development of PD symptoms, and the relative abundance of *Helicobacter* increased in AD mice ^{7,8,9}. *Alistipes* also the marker microbiota in model group, which was found to be an increasing abundance in the depression ¹⁰. Our results showed a decrease of *Helicobacter* and *Alistipes* in EE supplementation and FMT of EE (Fig.3G, 6E), which revealed that EE reduced the risk of cognitive impairment by regulating the intestinal microbiota.

6. It is not indicated whether bacteria that correlate with specific neurotransmitters are capable of synthesizing the neurotransmitters.

Answer:

Thank you very much for your valuable suggestions. Thank you very much for your valuable suggestions. We have re-written this part according to your suggestion.

Given that the microbiota has been proven to be able to synthesize and respond to several key neurochemicals (e.g., 5-HT, GABA, and others) that are involved in host mood, behavior, and cognition ¹¹, we found that the changes in these neurochemical signals do have a certain correlation with microbiota (Fig.5B). Furthermore, later fecal transplant experiments also verified this result (Fig.9A). *Ruminococcus* was involved in the synthesis of tryptamine, which induced the 5-HT

secretion¹². We found it was the marker bacteria in FMT_EE group. As studies have reported that *Streptococcus* is related to GABA synthesis¹³, in our study, correlation analysis showed that *Streptococcus* was related to GABA levels (Fig 5B).

7. Check English grammar and spelling problems.

Answer:

Thank you very much for your valuable suggestions. We have checked the English grammar through the manuscript and the modifications were marked in red.

Thank you for the reviewer's comments to give us an opportunity to think in-depth and improve our manuscript. We have systematically reviewed the full text and revised the language. We tried our best to improve the manuscript and made some changes in the manuscript. We hope that the correction will meet with approval.

Reviewer #2:

Therapeutic doses of radiation are often associated with a detrimental impact on brain function, often manifesting as cognitive dysfunction in a number of domains. Eleutheroside E (EE), an important active ingredient from *Acanthopanax senticosus*, is associated with protective behavioral and biochemical effects in a number of model systems following various insults. Although EE has anti-inflammatory properties, the mechanisms underpinning these observations remains unclear. In this study, the authors test the hypothesis that a four-week EE supplementation is protective against irradiation induced cognitive dysfunction by remodelling the gut microbiome. This is comprehensively assessed using a variety of approaches including fecal microbiota transplantation experiments. The main findings reported are that the potential therapeutic value of EE in radiation induced cognitive and memory impairments is regulated by the gut microbiota via neurotransmitter systems and PKA/CREB/BDNF signalling pathways.

This is an interesting study with a number of potentially important observations. I have a number of queries, suggestions and recommendations:

(1) From a conceptual perspective, the authors need to be much clearer about what hypothesis they are testing here. The introduction mentions mobile phone and space radiation for which is a pretty spurious association when they are really evaluating therapeutic doses of radiation. This needs to be explained in a much clearer and explicit way in the introduction.

Answer:

Thank you very much for your valuable suggestions. We are very sorry for my unclear explanation and inappropriate cite. Our laboratory has previously carried out a series of studies on simulated space radiation by ⁶⁰Co-γ rays. And we found that the doses used in this paper can simulate the biological effects of space radiation. Actually, we just simulated the biological effects of space radiation instead of space radiation dose. We have revised this part in the introduction and marked in red.

(2) The details supplied about the fecal microbiota transplant are very sparse and it is difficult to fully interpret until a more detailed account is provided. I would direct the authors to the following publication: Guidelines for reporting on animal fecal transplantation (GRAFT) studies: recommendations from a systematic review of murine transplantation protocols (<https://www.tandfonline.com/doi/full/10.1080/19490976.2021.1979878>). The information recommended in the checklist provided here should be included in a revised submission. In particular, it is important to know what steps, if any, were taken to preserve the viability of the microorganisms in the sample. It also appears as though the authors did not deplete the resident microbiota prior to the initiation of the microbiota transplant. Of further concern is that a critical control group is missing since if I understand correctly, it is only the EE treated group that act as donors. Without this group, it is difficult to know if there is a difference in behavioural and molecular profiles between the control and EE-remodelled microbiota.

Answer:

Thank you very much for your valuable suggestions. That helps us a lot. We supplemented relevant information of FMT with reference to GRAFT. And mark the revision in red. As mentioned in the references you provided, while antibiotic-induced depletion has, to date, been an area of critical methodological consideration in optimal FMT administration, it remains an area of contentious debate. In fact, increasing evidence suggests that antibiotic depletion may not be necessary for FMT uptake. In our study, we didn't used the antibiotic to delete the resident microbiota.

In addition, the control group in the FMT is very necessary. First of all, we are very sorry for our lack of thoughtfulness. Originally, we designed the control group in the fecal microbiota transplantation part, and evaluated the indicators involved in the article. The results of this part were analysis in another ongoing paper, so it is not reflected in this manuscript. It is our mistake. In order to ensure the integrity and scientificity of this manuscript, we have added this part to the revised manuscript and re-analyzed the results. Thanks again for your valuable comments. The changes were marked in red in the revised manuscript. (line77-85; line321-339; line347-349; line 353-356) And the Figure 6-Figure.9 have been updated.

Figure 6

Figure7

Figure 8

Figure 9

(3) Appropriate referencing is an issue throughout the manuscript. For example, authors cite a review paper about clinical FMT as the origin of the ‘established protocol’ they use in

their experiments. The papers cited in the introduction regarding radiation all appear to be in the context of therapeutic doses even though they are linked to mobile phone radiation.

Answer:

Thank you very much for your valuable suggestions. We rewritten this part of introduction and corrected the cite and FMT experiment were conducted following the reference you recommended us.

The fecal transplant was performed based on an established protocol¹⁶. Briefly, the donor mice of EE were administrated with EE for 4 weeks. And the control donors were administrated with saline for 4 weeks. Then, feces of the mice were collected daily and 100 mg was resuspended in 1 ml of sterile saline. The solution was mixed for 10 s using a vortex, before centrifugation at 800g for 3 min. The supernatant was collected and used as transplant material. Fresh transplant material was prepared on the same day of transplantation within 10 min before oral gavage to prevent changes in bacterial composition. The FMT mice were gavaged with the fresh transplant material daily for 4 weeks. After four weeks fecal transplantation, the mice were irradiated by ⁶⁰Co- γ ray at a total dose of 4 Gy.

(4) The methods section should provide the approval number of the ethics approval.

Answer:

Thank you very much for your valuable suggestions. We have provided the approval number of the ethics approval in the revised manuscript. All of the animal experimental procedures were approved by the local ethics committee of Harbin Institute of Technology (IACUC-2021075).

(5) The experimental details for the behavioural assays are also lacking in detail. What lighting conditions were used for the open field. Please also see this paper for the experimental parameters that should be reported for the novel object recognition (Object recognition test in mice <https://www.nature.com/articles/nprot.2013.155>).

Answer:

Thank you very much for your valuable suggestions. I have added the details of behavioural assays in the revised manuscript.

Novel object recognition

The novel object recognition test (NORT) was performed as previously reported with a little modified¹⁴. The experiment was carried out in a box (45×30×20cm, Fig.1B). The test was divided into habituation period, familiarization section and test section. The first and second days were the habituation period: each mouse was placed individually in the empty box for 5min. Day 3-4 is the familiarization period: put two identical objects in a symmetrical corner of the box. The mice were put into the box and their exploration time for the new and old two different objects were recorded within 5 minutes. The fifth day was the test period: one of the objects was replaced with a new object with the same position. The mice were put into the box and their exploration time for the new and old two different objects were recorded within 5 minutes. The discrimination index = $(T_n - T_0) / (T_n + T_0)$ The time exploring the new object is represented by T_n , whereas T_0 represents the time exploring the old object.

Open field test

For open field test, mice were placed separately in a corner of an open-field box (50 × 50 × 30 cm) with black vertical walls and the bottom was equally divided into 25 grids. The top of the box is open and illuminated by natural light. The mice were placed in the central grid of the open

field experiment box, and the vertical movement times of the rats and the activity time in the central area were observed and recorded.

(6) The authors have used β -actin as their housekeeper gene – did they verify that its expression was not altered by the experimental interventions?

Answer:

Thank you very much for your comments. According to the reference, β -actin was used as housekeeper gene, and radiation did not alter the expression of β -actin¹⁵. In our study, for the qPCR results, the expression of β -actin was not changed by the interventions.

(7) I'm not clear on the relevance or value of the correlations between ACH and GABA levels in brain and colon tissue. I would recommend removing this from a revised manuscript.

Answer:

Thank you very much for your valuable suggestions. I have removed this from the revised manuscript.

Lastly, thank you for the editor's and reviewer's comments which indeed positive and enlightened us to think more in-depth on our research. In all, we do hope that the responses and corrections can meet your approval.

Reference

1. Zhou Y, Cheng C, Baranenko D, Wang J, Li Y, Lu W. Effects of *Acanthopanax senticosus* on Brain Injury Induced by Simulated Spatial Radiation in Mouse Model Based on Pharmacokinetics and Comparative Proteomics. *Int J Mol Sci* **19**, (2018).
2. Zhou AY, *et al.* *Acanthopanax senticosus* reduces brain injury in mice exposed to low linear energy transfer radiation. *Biomed Pharmacother* **99**, 781-790 (2018).
3. Liu W-H, *et al.* Alteration of behavior and monoamine levels attributable to *Lactobacillus plantarum* PS128 in germ-free mice. *Behavioural Brain Research* **298**, 202-209 (2016).
4. Liang S, *et al.* Administration of *Lactobacillus helveticus* NS8 improves behavioral, cognitive, and biochemical aberrations caused by chronic restraint stress. *Neuroscience* **310**, 561-577 (2015).
5. Emge JR, *et al.* Modulation of the microbiota-gut-brain axis by probiotics in a murine model of inflammatory bowel disease. *American Journal of Physiology-Gastrointestinal and Liver Physiology* **310**, G989-G998 (2016).
6. Guo G, *et al.* Psychological stress enhances the colonization of the stomach by *Helicobacter pylori* in the BALB/c mouse. *Stress* **12**, 478-485 (2009).
7. Çamcı G, Oğuz S. Association between Parkinson's Disease and *Helicobacter Pylori*. *J Clin Neurol* **12**, 147-150 (2016).
8. Dobbs SM, *et al.* Peripheral aetiopathogenic drivers and mediators of Parkinson's disease and co-morbidities: role of gastrointestinal microbiota. *Journal of NeuroVirology* **22**, 22-32 (2016)
9. Bulgart HR, Neczypor EW, Wold LE, Mackos AR. Microbial involvement in Alzheimer

- disease development and progression. *Mol Neurodegener* **15**, 42 (2020).
10. Jiang H, *et al.* Altered fecal microbiota composition in patients with major depressive disorder. *Brain Behav Immun* **48**, 186-194 (2015).
 11. Lyte M. Microbial endocrinology in the microbiome-gut-brain axis: how bacterial production and utilization of neurochemicals influence behavior. *PLoS Pathog* **9**, e1003726 (2013).
 12. Williams BB, *et al.* Discovery and characterization of gut microbiota decarboxylases that can produce the neurotransmitter tryptamine. *Cell Host Microbe* **16**, 495-503 (2014).
 13. Yang SY, *et al.* Production of γ -aminobutyric acid by *Streptococcus salivarius* subsp. *thermophilus* Y2 under submerged fermentation. *Amino Acids* **34**, 473-478 (2008).
 14. Leger M, *et al.* Object recognition test in mice. *Nature Protocols* **8**, 2531-2537 (2013).
 15. Guo H, *et al.* Multi-omics analyses of radiation survivors identify radioprotective microbes and metabolites. *Science* **370**, (2020).

REVIEWERS' COMMENTS:

Reviewer #1 (Remarks to the Author):

I think the authors have revised the manuscript according to the comments. It is acceptable for the journal.

Reviewer #2 (Remarks to the Author):

All queries well addressed - I have no further comments.